# Implementation of an Analytical Method for Spectrophotometric Evaluation of Total Phenolic Content in Essential Oils

**DOI:** 10.3390/molecules27041345

**Published:** 2022-02-16

**Authors:** Delia Michiu, Maria-Ioana Socaciu, Melinda Fogarasi, Anamaria Mirela Jimborean, Floricuţa Ranga, Vlad Mureşan, Cristina Anamaria Semeniuc

**Affiliations:** 1Department of Food Engineering, University of Agricultural Sciences and Veterinary Medicine Cluj-Napoca, 3-5 Mănăştur Str., 400372 Cluj-Napoca, Romania; delia.michiu@usamvcluj.ro (D.M.); maria-ioana.socaciu@usamvcluj.ro (M.-I.S.); melinda.fogarasi@usamvcluj.ro (M.F.); vlad.muresan@usamvcluj.ro (V.M.); 2Department of Food Science, University of Agricultural Sciences and Veterinary Medicine Cluj-Napoca, 3-5 Mănăştur Str., 400372 Cluj-Napoca, Romania; floricutza_ro@yahoo.com

**Keywords:** total phenolic content, essential oils, Folin–Ciocalteu method, UV-Vis spectrophotometry, polyphenolic compounds, HPLC-DAD-ESI-MS analysis

## Abstract

Over the past decade, there has been growing interest in polyphenols’ research since these compounds, as antioxidants, have several health benefits, such as preventing neurodegenerative diseases, inflammation, cancer, cardiovascular diseases, and type 2 diabetes. This study implements an analytical method to assess the total phenolic content (TPC) in essential oils using Folin–Ciocalteu’s phenol reagent and quantifies the individual phenolic compounds by liquid chromatography. Thus, the research design and methodology included: (1) extraction of essential oil from dried thyme leaves by hydrodistillation; (2) spectrophotometric measurement of TPC by Folin–Ciocalteu method; and (3) identification and quantification of individual phenolic compounds by high-performance liquid chromatography-diode array detection/electrospray ionization mass spectrometry (HPLC-DAD-ESI-MS). Results revealed a TPC of 22.62 ± 0.482 mg GAE/100 µL and a polyphenolic profile characterized by phenolic acids (52.1%), flavonoids (16.1%), and other polyphenols (31.8%). Thymol, salvianolic acid A, and rosmarinic acid were the major compounds of thyme essential oil. The proposed analytical procedure has an acceptable level of repeatability, reproducibility, linearity, LOD (limit of detection), and LOQ (limit of quantification).

## 1. Introduction

Polyphenols are compounds found naturally in plants, as they are synthesized in their tissues, in fruits and vegetables [1]. Endogenous synthesis of phenolic compounds is due to plants’ response to ecological and physiological pressures such as pathogen and insect attacks, UV radiation, and wounding [2]. These compounds are recognized for their antioxidant properties, and their potential benefits have been largely studied [3,4,5]. The antioxidant activity of polyphenolic compounds is related to their reactive radical species inactivation capability; neutralization occurs when an antioxidant transfers its electron and/or hydrogen atom to the radical [6]. Previous studies have shown that many dietary polyphenolic constituents derived from plants are more effective antioxidants in vitro than vitamins E or C, and thus might contribute significantly to the protective effects in vivo [7]. Recent evidence has proposed that a high dietary intake of polyphenols may be inversely associated with overall and cardiovascular disease-related mortality, certain cancers, anthropometric measurements, and mood disorders [8].

Polyphenolic compounds are phytochemicals consisting of flavonoids and nonflavonoids, subdivided into multiple subclasses depending on the number of phenol units within their molecular structure, substituent groups, and/or the linkage type between phenol units [9]. Flavonoids are characterized by the presence of a 15-carbon (C6–C3–C6) backbone consisting of two phenyl units (A and B) and a heterocyclic unit (C) [10]. Structurally, they are grouped into subclasses, such as 2-phenylchromans (flavonoids, including flavanones, flavones, flavonols, flavanols, flavanonols, anthocyanins, and anthocyanidins) and 3-phenylchromans (isoflavonoids, including isoflavones, isoflavans, and pterocarpans) [9,11]. Non-flavonoid phenolic constituents include phenolic acids (hydroxycinnamic and hydroxybenzoic acids), volatile phenols, stilbenes (monomeric and oligomeric stilbenes), coumarins (simple coumarins, furanocoumarins, pyranocoumarins, and pyron-ring substituted coumarins), lignins (guaiacyl and guaiacyl-syringyl), lignans (furofuran, furan, dibenzylbutane, dibenzylbutyrolactone, aryltetralin, arylnaphthalene, dibenzocyclooctadiene, and dibenzylbutyrolactol), hydrolysable tannins (gallotannines, ellagitannines, and complex tannins), and condensed tannins (called procyanidins) [2,9,12,13,14,15,16].

Aromatic plants and their essential oils are rich in phenolic compounds, so-called phenylpropanoids [17]. While the main constituents of essential oils are generally mono- and sesquiterpenes, phenylpropanoid compounds are predominant in certain plant families. In addition, some terpenes are phenol derivatives, namely phenolic terpenes [18,19]. More recent evidence shows that the chemical composition of essential oils depends on several factors such as the domestication of plants, environmental conditions of plants growing, genetic differences, the plant part used for their isolation. Moreover, the harvesting season of plants and the extraction method, generally affect extraction yield and composition of essential oil [20,21].

The Folin–Ciocalteu method is the most used to assess total phenolic content (TPC) in essential oils. Its principle consists of phenolic compounds oxidation from the analyzed sample by the Folin–Ciocalteu reagent. This reagent is formed from a mixture of phosphotungstic acid (H_3_PW_12_O_40_) and phosphomolybdic acid (H_3_PMo_12_O_40_). After oxidation of the phenols, it is reduced to a mix of blue oxides of tungsten (W_8_O_23_) and molybdenum (Mo_8_O_23_). The blue coloration produced has maximum absorption in the region of 750 nm, and it is proportional to the total quantity of phenolic compounds initially present [22]. Gallic acid is usually used as the standard for quantitatively estimating phenolic compounds in the sample. Results are expressed as mg of gallic acid equivalents (GAE) per volume (mL or 100 µL) or mass (g) of essential oil.

There are a few studies in the literature on TPC measurement in thyme essential oil. For example, Lin et al. (2009) have investigated TPC levels in commercially available essential oils of *Thymus vulgaris* and *Thymus serpyllum* [23]. In addition, in 2014, Fatma et al. have measured TPC in essential oils of dried *Thymus hirtus* sp. *algeriensis* collected from different locations of Tunisia and in 2018, Aljabeili et al. in the essential oil of dried *Thymus vulgaris* from Egypt [24,25]. Furthermore, Mancini et al., in 2015, have assessed TPCs in essential oils of fresh *Thymus vulgaris* sampled from different areas of Italy [26]. More recently, Mutlu-Ingok et al. (2021) have quantified total phenolics in a commercial sample of thyme essential oil [27]. However, there is limited research on the utilization of liquid chromatography in evaluating polyphenolic profiles of thyme essential oils [24].

None of the above-mentioned reports gives a detailed description of the protocol followed to evaluate TPC in essential oil. In general, information about dilutions used, blank sample preparation, or standard concentrations is missing. Therefore, this study was designed as a detailed protocol for assessing TPC in essential oils, using the Folin–Ciocalteu method, to be implemented in an analytical laboratory. The experimental design included extraction of thyme essential oil by hydrodistillation, spectrophotometric evaluation of TPC, and determination of phenolic compounds profile in thyme essential oil by an HPLC-DAD-ESI-MS method. As far as we are aware, this is the first study demonstrating the suitability of this method for its intended use.

## 2. Results and Discussion

The extraction yield of thyme essential oil (2.2%) was calculated as the volume of essential oil (mL) per dried leaves’ weight (g) and multiplied by 100 [28].

Instrument calibration is an important stage in most measurement procedures. It involves (1) preparing a set of standards containing a known amount of the analyte of interest, the gallic acid in our case, (2) measuring the instrument response (absorbance value) for each standard, and (3) establishing the relationship between the instrument response and analyte concentration. This relationship is further used to transform measurements made on test samples into estimates of the amount of analyte present, as shown in Section 3.2.5. As calibration is a common and essential step in analytical methods, analysts must understand how to set up calibration experiments and evaluate results [29].

For constructing the calibration curve, five gallic acid standards were prepared in duplicate (in concentrations ranging from 0.25 to 1.25 mg/mL) and following the Folin-Ciocalteu method. Data from the calibration experiment were then plotted: on the *y*-axis, values of absorbances, and the *x*-axis, values of standards’ concentrations. Next, a linear regression through the origin was performed by using the five following data points: 1 (0.25, 0.187), 2 (0.50, 0.392), 3 (0.75, 0.613), 4 (1.0, 0.828), and 5 (1.25, 1.058), to establish the equation $\left(c=0.8323\times Abs.;\;R^{2}=0.9991\right)$ that best describes the linear relationship between absorbance value and gallic acid level, where *c* is the gallic acid concentration, 0.8323 is the slope of the regression line, and Abs. is the absorbance value at 750 nm. Finally, a regression analysis was carried out using the regression tool in Excel’s “Data Analysis ToolPak” to determine the correlation coefficient, ***r***, and significance of the correlation, ***p*-value**. The regression output created revealed several statistical parameters (see Table 1). The correlation coefficient (*r*), described as “Multiple R” in Excel output, indicates the strength and direction of the linear relationship between the two variables. It ranges from −1 (negative correlation) to +1 (positive correlation). When the ***r*-value** is in-between 0.00–0.19, the correlation is very weak; weak between 0.20 and 0.39, moderate between 0.40 and 0.59, strong between 0.60 and 0.79, and very strong between 0.80 and 1.0. The significance of the correlation is interpreted based on the *p*-value as follows: *p* ≥ 0.05^NS^, not significant trend; 0.01 ≤ *p* < 0.05, weakly significant; 0.005 ≤ *p* < 0.01, mildly significant; 0.001 ≤ *p* < 0.005, moderately significant; *p* < 0.001, strongly significant [30]. Considering the values resulting for *r* (0.999) and *p* (2.64 × 10^−12^), a very strong positive correlation was found between the two variables of the calibration curve, demonstrating its linearity within the range of the tested standards.

Limit of detection (LOD) and limit of quantification (LOQ) are two fundamental elements of method validation that define the limitations of an analytical method [31]. The LOD is the lowest amount of the analyte that can be detected by the method at a specified level of confidence. The LOQ is the lowest concentration of analyte that can be determined with an acceptable level of uncertainty and can, therefore, be set arbitrarily as the required lower end of the method working range [32].

To calculate the LOD (Equation (2)) and LOQ (Equation (3)) of this analytical procedure, first, we estimated the standard deviation (SD) of intercept with the following Equation (1):(1)SD of intercept=0.010778×√5
where 0.010778 is the intercept’s standard error (SE) (see the Table above) and 5 is the number of calibration standards.
(2)$$\mathrm{LOD}=3.3\times \frac{0.0241}{0.8323\;}$$
(3)$$\mathrm{LOQ}=10\times \frac{0.0241}{0.8323\;}$$
where 3.3 is the factor for LOD, 10 is the factor for LOQ, 0.0241 is the intercept’s standard deviation (SD), and 0.8323 is the slope of the calibration curve.

A LOD of 0.096 mg/mL and a LOQ of 0.290 mg/mL thus resulted. Therefore, even though the calibration curve is linear within the range of 0.25–1.25 mg GA/mL, phenolics can be quantified using this protocol up from a concentration of 0.29 mg GAE/mL oil.

Folin-Ciocalteu is a common method for assessing TPC, and many studies have reported this technique for phenolic compounds quantification in thyme essential oil. However, some necessary information in reproducing this method is missing, such as the sample dilution ratio. Since thyme essential oil has a content of phenolic compounds that exceeds the upper limit of quantification (ULOQ), to evaluate TPC, a dilution of the sample is needed. The upper limit of quantification (ULOQ) is the maximum analyte concentration that can be quantified with acceptable precision and accuracy (bias) in a sample. In general, the ULOQ is identical to the highest concentration of calibration curve [33]. Thus, when a sample is too concentrated, this must be diluted before analysis so that the analyte concentration falls in the middle of the calibration curve linear range [29]. Taking this into account, in the current research, we prepared a series of 16 dilutions of thyme essential oil in methanol (from 1:125 to 1:500) and subjected them to the Folin–Ciocalteu assay. Values of absorbances read at 750 nm were recorded in the worksheet from Table 2. The concentration of each dilution of thyme essential oil (x from the calibration curve), expressed in mg GAE/mL, was calculated by dividing the absorbance value read (y value) to 0.8323 (a value, the slope). As data are expressed as the mean value ± standard deviation (SD) of the three replicates, these values, including those of variation coefficients (CVs), were calculated for each dilution and recorded in the worksheet from Table 2. Generally, the CV, also known as relative standard deviation (RSD), a measure of precision between replicate measurements, should be <10% unless the method specifies otherwise [34]. Finally, the one-way analysis of variance (ANOVA) was performed, with Tukey’s honestly significant difference (HSD) post hoc test at *p* < 0.05, to determine statistically significant differences between these means. The concentration value closest toward the center of the calibration curve range belonged to the 1:300 thyme essential oil dilution (0.75 mg GAE/mL^h^). However, concentration values of 1:325 (0.76 mg GAE/mL^h^) and 1:350 (0.72 mg GAE/mL^hi^) dilutions of thyme essential oil were not significantly different from that of 1:300. Therefore, accurate results can be obtained in assessing the TPC from thyme essential oil when its dilutions are prepared in ratios between 1:300 and 1:350. To estimate TPC values in mg GAE/100 µL essential oil, the concentrations calculated using the calibration curve function were multiplied by the corresponding dilution factors then divided by 10. The level of TPC associated with the 1:300 dilution of thyme essential oil is 22.62 ± 0.482 mg GAE/100 µL (239.44 ± 5.099 mg GAE/g).

Significantly lower levels were found by Lin et al. (2009) in *Thymus vulgaris* essential oil (11.54 mg GAE/g) and *Thymus serpyllum* essential oil (55.10 mg GAE/g), by Fatma et al. (2014) in essential oils of *Thymus hirtus* sp. *algeriensis* collected from different locations of Tunisia (7.08–8.81 mg GAE/g), by Mancini et al. (2015) in essential oils of *Thymus vulgaris* sampled from different areas of Italy (77.6–165.1 mg GAE/g), and by Mutlu-Ingok et al. (2021) in the commercial sample of thyme essential oil (0.019 mg GAE/100 µL) [23,24,26,27]. However, comparable levels were reported by Aljabeili et al. (2018) in the Egyptian essential oil of *Thymus vulgaris* (177.3 mg GAE/g) [25].

Table 3 shows individual polyphenolic compounds quantified in the thyme essential oil taken in this study by HPLC-DAD-ESI-MS analysis. Eighteen compounds were identified in this essential oil and grouped into three chemical classes (phenolic acids-PAs, flavonoids-FVs, and other phenols-OPs) that included seven compound types (hydroxybenzoic acids-HBA, hydroxycinnamic acids-HCAs, flavanols-FVols, flavones-FVes, phenolic terpenes-PTs, caffeic acid derivatives-CADs, and coumaric acid derivatives-CODs). The most abundant constituents were phenolic acids [52.1% (4.5% HBAs, 18.9% HCAs, 21.9% CADs, and 6.7% CODs), followed by other polyphenols [31.8%, represented by PTs], and flavonoids [16.1% (2.0% FVols and 14.0% FVes)]. The major compounds in thyme essential oil were the thymol (333.37 µg/mL), a phenolic terpene, salvianolic acid A (223.33 µg/mL), a caffeic acid derivative, and rosmarinic acid (114.73 µg/mL), an hydroxycinnamic acid.

Although the volatile composition of thyme essential oil by gas-chromatography is well-studied, there is a lack of studies on its polyphenolic composition using liquid chromatography. Our literature review revealed two such studies, the one of Hajimehdipoor et al. (2010) on essential oil of *Thymus vulgaris* L. [35], and the one of Fatma et al. (2014) on essential oil of *Thymus hirtus* sp. *algeriensis* [24]. Thymol was also the majority compound in thyme essential oil studied by Hajimehdipoor et al. (2010), followed by carvacrol [35]. However, Fatma et al. (2014) reported a different polyphenolic profile of their thyme essential oil, with hydroxybis, tyrosin, and hydroxyphenyl acetic acid as main compounds [24].

## 3. Materials and Methods

### 3.1. Plant Material and Extraction of Thyme Essential Oil

For essential oil extraction, dried thyme leaves of *Thymus vulgaris* were purchased from a company that markets food ingredients (Solina Group, Alba Iulia, Romania http://www.solina-group.ro; accessed on 24 January 2022). Thyme essential oil was obtained by hydrodistillation using a Clevenger-type apparatus (S.C. Energo-Metr S.R.L., Odorheiu Secuiesc, Romania); 50 g of thyme leaves were boiled, for 3 h, in 750 mL distilled water. The collected essential oil was dried over sodium sulphate anhydrous and stored at 4 °C until analysis.

### 3.2. Spectrophotometric Evaluation of Total Phenolic Content (TPC) in Essential Oil by Folin-Ciocalteu Method

The operating procedure proposed in this paper is a deliverable resulting from implementing this method in our research laboratory [19]. Step-by-step instructions are below detailed.

#### 3.2.1. Scope

The present operating procedure specifies a method for assessing TPC in essential oil using the Folin–Ciocalteu method. This method is suitable for essential oil having a TPC of up to 42 mg gallic acid equivalent (GAE)/100 µL.

#### 3.2.2. Principle

A test portion (100 µL) of diluted essential oil is dissolved in distilled water (6.0 mL). Next, 2 N Folin–Ciocalteu’s phenol reagent (0.5 mL), 0.71 M sodium carbonate solution (1.5 mL), and the rest of distilled water (1.9 mL) are added to reach a 10 mL volume. After a fixed period of reaction (2 h), the content of phenols is calculated from a photometric measurement of the blue mixture of tungsten (W_8_O_23_) and molybdenum (Mo_8_O_23_) oxides.

#### 3.2.3. Preparation of Working and Standard Solutions

Only reagents of recognized analytical grade and distilled water were used.

Reagents:Gallic acid (2699.1, Carl Roth GmbH & Co. KG, Karlsruhe, Germany)Folin–Ciocalteu’s phenol reagent (F9252, Sigma-Aldrich Co., Saint Louis, MO, USA)Sodium carbonate anhydrous (27767.295, VWR, Leuven, Belgium)

Preparation of 7.5% (*m*/*v*) sodium carbonate solution [0.71 M sodium carbonate solution]. Weigh 7.5 g of sodium carbonate anhydrous into a 100 mL volumetric flask and make up to the mark with distilled water.

Preparation of 2.5 mg/mL gallic acid stock solution. Weigh 0.250 g of gallic acid into a 100 mL volumetric flask and make up to the mark with distilled water.

Preparation of gallic acid standard solutions (in a range from 0.25 mg/mL to 1.25 mg/mL). Five gallic acid standard solutions (Std. sol. I-0.25 mg/mL, Std. sol. II-0.50 mg/mL, Std. sol. III-0.75 mg/mL, Std. sol. IV-1.00 mg/mL, and Std. sol. V-1.25 mg/mL) are prepared in 50 mL volumetric flasks using the gallic acid stock solution and distilled water in different ratios. Prepare all gallic acid standard solutions in duplicate.

#### 3.2.4. Apparatus and Accessories

Analytical balance (ABJ-220-4NM; Kern & Sohn GmbH, Balingen, Germany)Vortex (Vortex V-1 Plus; Biosan Ltd., Riga, Latvia)Double beam UV-VIS spectrophotometer (UV-1900i; Shimadzu Scientific Instruments, Inc., Columbia, MD, USA)Quartz cuvettes with lid, 1.4 mL, 10 mm path (Semi-Micro Cell 104-QS; Hellma Analytics, Munich, Germany)

#### 3.2.5. Working Procedure for Preparation of Calibration Curve

Preparation of Blank Sample

Transfer 100 µL of methanol into a 16-mL glass bottle with a rubber stopper.Add 6 mL distilled water and 0.5 mL of 2 N Folin–Ciocalteu’s phenol reagent and mix using the vortex.After 4 min, add 1.5 mL of 0.71 M sodium carbonate solution and 1.9 mL distilled water and vortex again.Keep the mixture in the dark, at room temperature, for 2 h.Turn on the spectrophotometer.Set the wavelength of the spectrophotometer to 750 nm.Transfer approximately 1.4 mL of the blank sample into each quartz cuvette.Insert the quartz cuvettes into the spectrophotometer.Press the “autozero” key.

Measurement of Standards (see Figure 1) for Calibration Curve Preparation

Transfer 100 µL of the gallic acid standard solution into a 16-mL glass bottle with a rubber stopper.Add 6 mL distilled water and 0.5 mL of 2 N Folin–Ciocalteu’s phenol reagent and mix using the vortex.After 4 min, add 1.5 mL of 0.71 M sodium carbonate solution and 1.9 mL distilled water and vortex again.Keep the mixture in the dark, at room temperature, for 2 h.Transfer approximately 1.4 mL of the standard into the quartz cuvette from the front rack.Read and record the absorbance value of the standard (see Table 4).Repeat all above-mentioned steps for each standard.Generate the gallic acid calibration curve in Microsoft Excel by plotting values of standards’ absorbances (Oy) vs their concentrations (Ox) (see Figure 2).

#### 3.2.6. Working Procedure for Measurement of TPC in Essential Oil

Preparation of Blank Sample

Transfer 100 µL of methanol into a 16-mL glass bottle with a rubber stopper.Add 6 mL distilled water and 0.5 mL of 2 N Folin-Ciocalteu’s phenol reagent and mix using the vortex.After 4 min, add 1.5 mL of 0.71 M sodium carbonate solution and 1.9 mL distilled water and vortex again.Keep the mixture in the dark, at room temperature, for 2 h.Turn on the spectrophotometer.Set the wavelength of the spectrophotometer to 750 nm.Transfer approximately 1.4 mL of the blank sample into each quartz cuvette.Insert the quartz cuvettes into the spectrophotometer.Press the “autozero” key.

Measurement of Test Sample

Transfer 100 µL of the prepared test sample [thyme essential oil diluted in methanol (20847.320P, VWR Chemicals, Fontenay-sous-Bois, France)-see Table 5] into a 16-mL glass bottle with a rubber stopper.Add 6 mL distilled water and 0.5 mL of 2 N Folin–Ciocalteu’s phenol reagent and mix using the vortex.After 4 min, add 1.5 mL of 0.71 M sodium carbonate solution and 1.9 mL distilled water and vortex again.Keep the mixture in the dark, at room temperature, for 2 h.Transfer approximately 1.4 mL of the test sample into the quartz cuvette from the front rack.Read and record the absorbance value of the test sample.Calculate TPC using the equation from Section 3.2.4 and record the value.
○Calculate the mean of the three values using the AVERAGE function in Excel.○Calculate the standard deviation of the three values using the STDEV function in Excel.○Calculate the coefficient of variation (CV) by multiplying the ratio of standard deviation to mean by 100.Report results as mean ± standard deviation of the three measurements.

#### 3.2.7. Calculation and Expression of Results

Calculate the total phenolic content (TPC), expressed as mg gallic acid equivalent (GAE)/100 µL essential oil, using the following Equation (4):(4)$$\mathrm{TPC}=\frac{\frac{\mathrm{Abs}.-b}{m}\times d}{10}$$
where Abs. is the absorbance value at 750 nm, b is the y-intercept of the linear equation, m is the slope of the regression line, and d is the test sample dilution factor.

### 3.3. Identification and Quantification of Phenolic Compounds in Thyme Essential Oil by HPLC-DAD-ESI-MS

#### 3.3.1. Preparation of Thyme Essential Oil Methanolic Extract

It was carried out using the protocol described by Ricciutelli et al. (2017) [36], with minor modifications. An aliquot (3 mL; ~2.83 g) of thyme essential oil was transferred into a 15-mL Falcon centrifuge tube; 6 mL of *n*-hexane was added and vortexed (V-1 Plus vortex; Biosan Ltd., Riga, Latvia) for 30 s, then 5 mL of methanol/water (3:2, *v*/*v*) and sonicated for 15 min (Sonorex RK 100 H ultrasonic bath; Bandelin electronic GmbH & Co. KG, Berlin, Germany). The mixture was then centrifuged at 867× *g* (3,000 rpm) for 10 min (EBA 20 centrifuge; Andreas Hettich GmbH & Co. KG, Tuttlingen, Germany), and the lower phase was collected (methanol-aqueous phase). Next, 3 mL of *n*-hexane was added to the methanol-aqueous extract and vortexing, sonication, centrifugation, and phase separation steps were repeated until a clear extract was obtained (three times). Finally, the solvent was removed by evaporation at 40 °C (Hei-VAP Expert rotary evaporator; Heidolph Instruments GmbH & Co. KG, Schwabach, Germany), and the dry extract was resuspended in 1 mL of methanol. The obtained methanolic extract was filtered through a polyamide syringe filter (0.45 µm pore size, 25 mm diameter) then stored at −18 °C until chromatographic analysis.

Polyamide syringe filters (Chromafil Xtra PA 45/25; Macherey-Nagel GmbH & Co. KG, Düren, Germany), *n*-hexane (Sigma-Aldrich Co., Saint Louis, MO, USA), and methanol, LC-MS grade (Supelco, Inc., Bellefonte, PA, USA) were used as materials and reagents for the extraction of polyphenolic compounds from thyme essential oil.

#### 3.3.2. HPLC-DAD-ESI-MS Analysis of Thyme Essential Oil Methanolic Extract

It was performed using the method published by Fogarasi et al. (2021) [30]. Separation, identification, and quantification of individual phenolic compounds were performed on a liquid chromatography system (1200 HPLC; Agilent Technologies Inc., Palo Alto, CA, USA). It contained a photodiode array (PDA) detector (G1315B), a single-quadrupole mass spectrometer (MS) (G6110) equipped with an electrospray ionization (ESI) source (G1948B). The system also included a quaternary pump (G1311A), a degasser (G1322A), an autosampler (G1329A), a thermostatted column compartment (G1316A), a Kinetex XB-C18 column (150 mm L × 4.6 mm ID × 5 μm particle size; Phenomenex, Torrance, CA, USA), and the ChemStation software (Rev B.04.02 SP1; Agilent Technologies Inc., Palo Alto, CA, USA).

Twenty microliters of thyme essential oil extract were injected into the HPLC system for analysis. In the elution process, two mobile phases were employed: solution A containing 0.1% acetic acid in ultrapure water and solution B containing 0.1% acetic acid in acetonitrile. A multi-step gradient elution model was used with the following settings: 5% B (0–2 min), from 5 to 40% B (2–18 min), from 40 to 90% B (18–20 min), 90% B (20–24 min), from 90 to 5% B (24–25 min), 5% B (25–30 min). The flow rate was programmed to 0.5 mL/min and the column oven temperature to 25 °C. The PDA was set to scan from 200 to 600 nm. The chromatograms were acquired at 280; data acquisition was performed for 30 min.

Mass spectra were recorded using the ESI source set in positive-ion mode and the MS in full scan mode (in an *m*/*z* range of 120−1200). Nitrogen was used as a drying gas. Other settings for the ESI source were as follows: drying gas temperature 350 °C, drying gas flow rate 7 L/min, nebulizer pressure 35 psi, capillary voltage 3000 V, fragmentor voltage 100 eV.

Phenolic compounds were tentatively identified by comparing their retention times, UV-vis spectra, and mass spectra to those of the standards analyzed under the same conditions and data available in the literature [37].

Polyamide syringe filters (Chromafil Xtra PA 45/25; Macherey-Nagel GmbH & Co. KG, Düren, Germany), glacial acetic acid, LC-MS grade (Supelco Inc., Bellefonte, PA, USA), ultrapure water (Supelco Inc., Bellefonte, PA, USA), acetonitrile, LC-MS grade (Supelco Inc., Bellefonte, PA, USA), chlorogenic acid (Phytolab GmbH & Co. KG, Vestenbergsgreuth, Germany), gallic acid (Phytolab GmbH & Co. KG, Vestenbergsgreuth, Germany), luteolin (Phytolab GmbH & Co. KG, Vestenbergsgreuth, Germany), methanol, LC-MS grade (Supelco Inc., Bellefonte, PA, USA), and nitrogen (SIAD România s.r.l., Bucharest, Romania) were used as materials and reagents for the quantification of polyphenolic compounds by HPLC-DAD-ESI-MS.

Standards of gallic acid, chlorogenic acid, and luteolin were prepared with methanol as solvent. Hydroxybenzoic acids, terpenes, and stilbenes were quantified using a five-point analytical curve of gallic acid (5−100 µg/mL; *r*^2^ = 0.9978); hydroxycinnamic acids using a five-point analytical curve of chlorogenic acid (10−50 µg/mL; *r*^2^ = 0.9937); flavones and flavanones using a five-point analytical curve of luteolin (0−100 µg/mL; *r*^2^ = 0.9972). The limit of detection [LOD = 3.3 × (Sy-the standard deviation of the response/S-the slope of the calibration curve)] was 0.35 μg/mL, and the limit of quantification [LOQ = 10 × (Sy/S)] was 1.05 μg/mL in the case of gallic acid; 0.41 μg/mL LOD and 1.64 μg/mL LOQ in case of chlorogenic acid; 0.21 μg/mL LOD and 0.84 μg/mL LOQ in case of luteolin. Samples were tested in triplicate. Results were expressed in µg/mL essential oil.

#### 3.3.3. Statistical Analysis

For data analysis, Minitab statistical software (version 19.1.1; LEAD Technologies, Inc., Charlotte, NC, USA) was used. The difference between values of dilutions was determined using one-way ANOVA. Post-hoc pairwise comparisons were performed with Tukey’s test at a 95% confidence level (*p* < 0.05).

## 4. Conclusions

This paper provides a laboratory procedure for spectrophotometric evaluation of TPC in thyme essential oil, containing instructions for conducting a calibration experiment and explanations on how to evaluate results. All analytical parameters analyzed showed adequate results after implementing the conditions for colorimetric assessment of phenolic compounds by the Folin-Ciocalteu’s phenol reagent. Furthermore, given the details provided by this procedure, researchers can easily replicate or use then in their experimental laboratory investigations. Moreover, they can routinely use the proposed protocol to assess TPC in any methanolic extract.

## Figures and Tables

**Figure 1 molecules-27-01345-f001:**
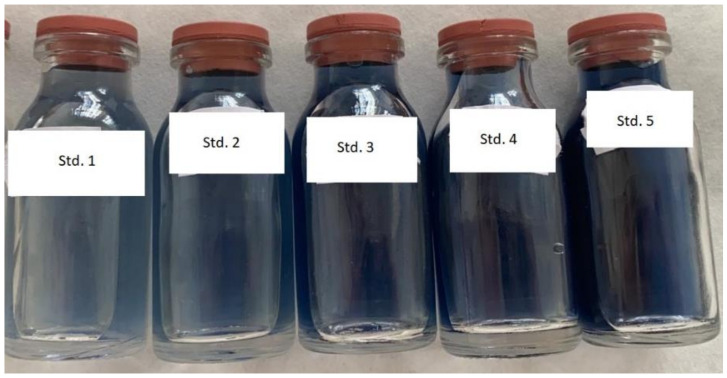
Visualization of standard solutions.

**Figure 2 molecules-27-01345-f002:**
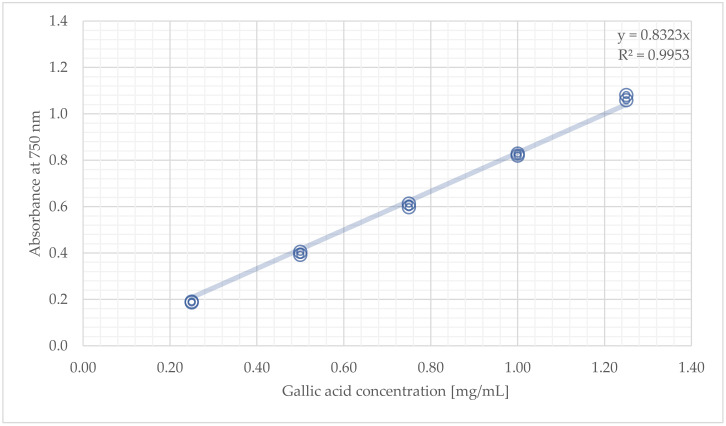
Gallic acid calibration curve for TPC evaluation.

**Table 1 molecules-27-01345-t001:** Statistics of the calibration curve.

**Regression Statistics**								
Multiple R						0.999118		
R Square						0.998238		
Adjusted R Square						0.998017		
Standard Error						0.014533		
Observations						10		
**ANOVA**								
	**df**	**SS**	**MS**		**F**		**Significance F**	
Regression	1	0.957031	0.957031		4531.264		2.64 × 10^−12^	
Residual	8	0.00169	0.000211					
Total	9	0.958721						
	**Coefficients**	**Standard Error**	**t Stat**	** *p* ** **-value**	**Lower 95%**	**Upper 95%**	**Lower 95%**	**Upper 95%**
Intercept	−0.03915	0.010778	−3.63243	0.006662	−0.064	−0.0143	−0.064	−0.0143
X Variable 1	0.875	0.012999	67.31466	2.64×10^−12^	0.845025	0.904975	0.845025	0.904975

**Table 2 molecules-27-01345-t002:** Worksheet for recording values of absorbances corresponding essential oil samples.

TEO Sample	Dilution Factor	Absorbance Value at 750 nm			TPC [mg GAE/100 µL]		
		Mean	SD	CV	Mean	SD	CV
Dilution of 1:500	501	0.57 ^k^	0.011	2.0	28.39	0.559	2.0
Dilution of 1:475	476	0.56 ^k^	0.008	1.4	26.82	0.375	1.4
Dilution of 1:450	451	0.57 ^k^	0.019	3.3	25.65	0.850	3.3
Dilution of 1:425	426	0.66 ^j^	0.010	1.6	27.91	0.447	1.6
Dilution of 1:400	401	0.71 ^i^	0.009	1.3	28.28	0.376	1.3
Dilution of 1:375	376	0.70 ^i^	0.011	1.6	26.47	0.431	1.6
Dilution of 1:350	351	0.72 ^hi^	0.008	1.2	25.30	0.292	1.2
Dilution of 1:325	326	0.76 ^h^	0.005	0.7	24.73	0.163	0.7
Dilution of 1:300	301	0.75 ^h^	0.016	2.1	22.62	0.482	2.1
Dilution of 1:275	276	0.84 ^g^	0.003	0.4	23.12	0.083	0.4
Dilution of 1:250	251	0.92 ^f^	0.023	2.5	22.98	0.575	2.5
Dilution of 1:225	226	1.04 ^e^	0.032	3.1	23.57	0.729	3.1
Dilution of 1:200	201	1.18 ^d^	0.017	1.4	23.74	0.342	1.4
Dilution of 1:175	176	1.23 ^c^	0.021	1.7	21.63	0.362	1.7
Dilution of 1:150	151	1.63 ^b^	0.014	0.8	24.63	0.209	0.8
Dilution of 1:125	126	3.41 ^a^	0.0	0.0	42.90	0.0	0.0

TEO—thyme essential oil; SD—standard deviation; CV—coefficient of variation. Different letters in the same row indicate a statistically significant difference at *p* < 0.05 (Tukey’s test).

**Table 3 molecules-27-01345-t003:** Content of polyphenolic compounds (µg/mL) in thyme essential oil of the current study and comparison with literature data.

Crt. No.	Compound	Chemical Class	Type of Compound	TEO (µg/mL)	Literature Data		
					Studied *Thymus* Genotypes	Analytical Method/Content	Ref.
1	Gallic acid	PAs	HBA	3.40 ± 0.211	*Thymus hirtus* sp. *algeriensis*	HPLC-DAD/433.92 μg/g dw	[24]
2	*p*-Hydroxybenzoic acid	PAs	HBA	65.14 ± 0.378	-	-	-
3	Caffeic acid	PAs	HCA	49.80 ± 0.313	-	-	-
4	Epicatechin	FVs	FVol	30.45 ± 0.313	--		-
5	*p*-Coumaric acid	PAs	HCA	62.91 ± 0.375	-	-	-
6	Ferulic acid	PAs	HCA	58.01 ± 2.473	*Thymus hirtus* sp. *algeriensis*	HPLC-DAD/433.92 μg/g dw	[24]
7	Apigenin-7-*O*-glucoside	FVs	FVe	36.85 ± 0.976	-	-	-
8	Luteolin-7-*O*-glucoside	FVs	FVe	66.93 ± 4.036	-	-	-
9	Rosmarinic acid	PAs	HCA	114.73 ± 2.849	-	-	-
10	Naringenin	FVs	FVe	35.26 ± 1.889	-	-	-
11	Apigenin	FVs	FVe	47.29 ± 0.585	-	-	-
12	Luteolin	FVs	FVe	25.68 ± 1.335	-	-	-
13	Methyl rosmarinate	PAs	COD	101.91 ± 2.192	-	-	-
14	Rosmadial	OPs	PT	92.65 ± 2.313	-	-	-
15	Salvianolic acid C	PAs	CAD	107.83 ± 2.334	-	-	-
16	Salvianolic acid A	PAs	CAD	223.33 ± 21.451	-	-	-
17	Carvacrol	OPs	PT	55.03 ± 0.0	*Thymus vulgaris* L.	HPLC-DAD/4.3% (*w*/*w*)	[35]
18	Thymol	OPs	PT	333.37 ± 42.480	*Thymus vulgaris* L.	HPLC-DAD/40.4% (*w*/*w*)	[35]
**Total content**				**1510.58**	-	-	-

TEO—thyme essential oil; PAs—phenolic acids; FVs—flavonoids; OPs—other polyphenols; HBA—hydroxybenzoic acid; HCA—hydroxycinnamic acid; FVol—flavanol; FVe—flavone; PT—phenolic terpene; CAD—caffeic acid derivative; COD—coumaric acid derivative; Ref.—bibliographic reference. Values are expressed as mean ± standard deviation of three replicates.

**Table 4 molecules-27-01345-t004:** Worksheet for recording values of absorbances corresponding standards.

Standard	Repetition	Absorbance Value at 750 nm
Std. 1	1 2	0.190
		0.187
Std. 2	1 2	0.405
		0.392
Std. 3	1 2	0.597
		0.613
Std. 4	1 2	0.820
		0.828
Std. 5	1	1.058
	2	1.081

**Table 5 molecules-27-01345-t005:** Preparation of thyme essential oil dilutions.

Dilution of TEO	TEO [µL]	Methanol [mL]
1:25	50	1.25
1:50	50	2.50
1:75	50	3.75
1:100	50	5.00
1:125	50	6.25
1:150	50	7.50
1:175	50	8.75
1:200	50	10.00
1:225	50	11.25
1:250	50	12.50
1:275	50	13.75
1:300	50	15.00
1:325	40	13.00
1:350	40	14.00
1:375	40	15.00
1:400	30	12.00
1:425	30	12.75
1:450	30	13.50
1:475	30	14.25
1:500	30	15.00

TEO—thyme essential oil.

## Data Availability

Not applicable.

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
