# Peer review of "Implementation of an Analytical Method for Spectrophotometric Evaluation of Total Phenolic Content in Essential Oils"

_molecules, 2022, doi:10.3390/molecules27041345_

Round 1

Reviewer 1 Report

Dear Authors, 

This is an important submission toward Molecules. However, it needs major revisions to be recommended for publication within Molecules. My comments are included in the attached file (the original submission PDF file). Please note that English should be revised.

Kind regards.     

Author Response

We thank the reviewer for the careful reading of the manuscript and especially for his/her helpful comments. We have revised the manuscript according to your comments point by point. Please see the revised manuscript and the following answers. All changes are highlighted with track changes in the revised manuscript.

We also like to mention that a native English speaker reviewed the manuscript.

Reviewer 2 Report

Dear sirs,

the work is interesting, yet it's not particularly innovative.
I found minor spell errors in writing formulas (rows 77-79) Please use subscript for the numbers in formulas.

I found a principal issue on Figure 2 and then a methodological/conceptual explanation is required.
I would kindly ask why the regression line do not pass through the origin of the axes.

If you autozeroed the instrument with the blank reaction, then 0 abs = 0 concentration.
That implies the discussion and part of results should be modified. please explain.
Thank you

Author Response

We thank the reviewer for the careful reading of the manuscript and especially for his/her helpful comments. We have revised the manuscript according to your comments point by point. Please see the revised manuscript and the following answers. All changes are highlighted with track changes in the revised manuscript.

Round 2

Reviewer 1 Report

Dear Authors, 

It seems that the manuscript has greatly improved and therefore I suggest its publication in Molecules. 

Regards.